# Rainfall Flooding in Urban Areas in the Context of Geomorphological Aspects

**Tomasz Walczykiewicz *** and **Magdalena Skonieczna**

Institute of Meteorology and Water Management—National Research Institute, 01-673 Warsaw, Poland;
magdalena.skonieczna@imgw.pl
* Correspondence: tomasz.walczykiewicz@imgw.pl

**Abstract:** Flooding risk in urban areas is particularly high, due to the high population density and property values, including those of transport, residential, service and industrial infrastructure, among others. There are many reasons for flooding in urban areas; among them, direct heavy rainfall can cause special problems in risk management. In the case of random heavy rainfall, flood risk management can be supported by information about the morphology of the terrain and the degree of its sealing. In this study, we analyse methods for determining the risk of flooding in urban areas using digital terrain model (DTM) and geographic information system (GIS) tools. Predictors of precipitation floods in urban areas are defined, including the determination of flat areas, areas without outflow (non-drainage) and with large terrain height differences. The main source of information about historical rainfall floods relates to interventions by fire brigades, which constitute the basis for verifying the areas of occurrence of rainfall floods, as determined on the basis of morphological analysis of the area. Identifying the locations of rainfall flooding areas and developing accurate maps based on them are crucial for spatial planning and flood management at the local scale.

**Keywords:** flood risk; cloudburst; urban flood hazard mapping

## 1. Introduction

Geomorphological conditions have a significant impact on land development. They define the limits of the possibility of surface exploitation through both the morphological and morphometric features of the relief forms and the risk of erosion and accumulation processes [1]. In recent years, a characteristic phenomenon in urbanized areas is urban flooding, which often occurs in places with high rainfall or in adjacent lower areas. This type of flood often occurs without any element of the hydrographic network, sometimes in places where no element of the network has ever existed. The causes of this phenomenon are related not only to the occurrence of rain at a specific capacity or intensity but, above all, to the progressive sealing of urbanized surfaces [2,3] and the topography.

As a result of intense rainfall or cloudburst, violent flood waves can form in small catchments. In built-up areas, as a result of floods, these can cause significant material losses which, according to research conducted in Poland, may reach up to 30,000 euro per 1 km$^2$ of catchment area [4]. At present, the progressing urbanization of large cities has densified the development so much that the water retention capacity of these areas has decreased and, in many cases, has significantly decreased. In these cases, the effective precipitation, which is understood as the fraction of the total precipitation which remains after subtracting the losses related to interception, terrain wetting, filling of land depressions, infiltration, and evaporation [5,6], is almost equal to the actual precipitation.

Due to such losses, including losses resulting from floods, many international initiatives have been devoted to cities. These include Goal 11 among the Sustainable Development Goals [7]. The World Meteorological Organization has also drawn attention to the problems of extreme meteorological and

hydrological phenomena in cities. One of the results of the 18th World Meteorological Congress is highlighting the problem of urbanization [8]. It is also important to look for solutions in the field of integrated meteorological and environmental services for cities as critical areas [9]. In addition to numerous hazards and the growing risk of anomalous and dangerous situations, the threat of an increasing domino effect in urban areas has begun to emerge: A single extreme phenomenon may lead to new hazards and, ultimately, to widespread failure of the city's infrastructure [10].

Urban floods pose the greatest threat to areas with high development intensity. These areas are particularly exposed to local flooding and increases in surface runoff, which most often result in the destruction of property, buildings, and communication routes. In addition to the most obvious effects of urban floods, cities also have to deal with negative effects in other urban sectors; for example, the water and wastewater sectors are at risk of both inefficiency (or lack) of the drainage system and limited possibilities of producing and distributing treated water.

The Floods Directive defines flooding as "the temporary covering of land that is not normally covered with water by water, which includes floods caused by rivers, mountain streams, Mediterranean seasonal watercourses and coastal storm floods, but may not take into account floods caused by sewerage systems" [11]. The above definition does not exclude rainfall floods in urbanized areas. The definition of rainfall floods has been included in the Flood Directive Reporting Guidance [12]. Rainfall floods are associated with flooding of an area by water coming directly from rainfall or melting snow, which includes both urban storm floods and excess water in non-urban areas.

Urban floods are most often defined as events caused by heavy rainfall (above 20–25 mm/h) of short duration [13]. Sometimes floods of this type occur with lower intensity (about 10 mm/h) but longer duration, especially if the ground surface is poorly permeable (e.g., frozen or saturated with water) [14], or if the municipal sewage systems are overloaded. Then, the rain turns into a surface runoff, which quickly moves to the lowest points in the terrain and creates temporary flooding. Of course, urban floods should be distinguished from flash floods. The only (or, rather, the main) element connecting these events is the short time duration of flooding. However, their course, effects, and mitigation methods are significantly different. The course of a flash flood is, in practice, the rapid runoff of a large mass of water, which poses a threat to human life. The greatest number of people die during such events. They are related to rivers, although floods due to failures of dikes or dams are also considered to fall into this category. Rainfall floods, in turn, are the result of sudden and short-term rainfall and do not pose a significant threat to human life. However, they can significantly disrupt the functioning of the urban structure.

Research [15] has shownthat, in countries in the European Union, the percentage of people living in cities and suburbs is higher than in the rest of the world. According to LUISA [16] forecasts, this percentage will continue to increase until 2030 and reach a stable state by 2050. By 2050, the percentage of the population living in functional urban areas (i.e., cities and connected areas) is expected to be 70% of the total population of the EU. The problem of urban floods, therefore, will increase. The World Economic Forum's risk perception study showed that two-thirds of the world's population will live in cities by 2050, and an estimated 800 million people living in more than 570 coastal cities will be exposed to 0.5 m sea-level rise by 2050 [17]. Many cities in Europe and in the world, due to the increase in the number of inhabitants, changes in city structures, and the sealing of surfaces, face problems relating to urban floods. Significant increases in human settlements, especially in densely populated urban areas, flood plains and coastal zones, further deteriorate public exposure and vulnerability to hydrometeorological hazards due to climate variability. In 2017, the World Economic Forum identified extreme weather conditions and natural disasters as the greatest risk, with the cost of natural disasters in 2016 alone increasing to US \$175 billion, well above the 1986 – 2015 average of US \$126 billion [18].

Flood management is directly related to the identification of at risk areas. Designating urban areas which are at risk of rainfall flooding is a task in which, due to the specificity of the phenomenon, different methodological solutions should be used than those in the case of a hazard caused by rivers. In the case of rivers, for the delimitation of such areas, values of the flows with a certain excess value

are used to determine the extent of the risk area [19,20]. The size of the affected area also depends on the shape of the river valley. In the case of rainfall floods, the randomness of precipitation does not allow for its direct use in determining at risk areas. Therefore, it is necessary to look for other methods, such as those based on geomorphological analyses, allowing for their designation. In the case of urbanized areas, their designation is important in planning possible rescue actions, planning evacuation routes, and prioritizing these actions, which may constitute the basis for recommendations in the field of spatial planning and investment activities to modify the existing spatial development.

Thanks to such analyses, it becomes possible to identify areas which are susceptible to rainfall floods, due to the morphology of the land and its development. In conjunction with local rainfall monitoring systems operated by municipal services, a significant contribution can be made to the municipal flood management system [9].

## 2. Materials and Methods

### 2.1. Analysis of the Problem of Rainfall Floods in the Research Area - Questionnaire Research

During the research carried out by the authors [19], before the publication of the first flood risk management plans, municipalities were surveyed in terms of flood hazard, in order to identify its main sources. Among the endangered municipalities, more than a half indicated water runoff on the ground surface caused by rainfall and rising groundwater levels as the main cause of losses: 63% (469) and 61% (456) municipalities, respectively. The cause of "accumulation of water from rainfall in non-drainage areas" was also high in the ranking, as indicated by 285 (38%) municipalities. "Relatively slow rise in the water level in the river" and "rapid river floods" were indicated by 45% (334) and 38% (282) municipalities, respectively. Figure 1 presents the questionnaire results.

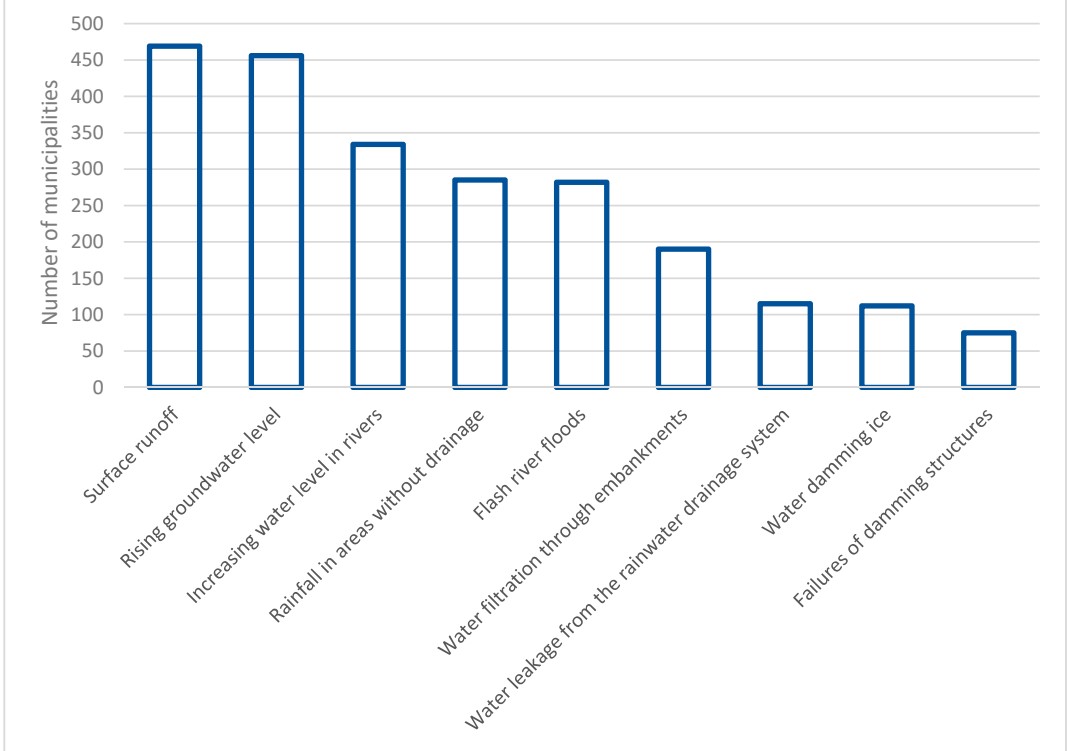

**Figure 1.** The causes of floods in Poland. Results of a survey of municipalities, own study based on Malopolska Grupa Geodezyjno-Projektowa (MGGP SA), Institute of Meteorology and Water Management-National Research Institute (IMWM-NRI); Analysis of the current flood protection system for the development of flood risk management plans for river basins and water regions (unpublished) [21].

Analysis of responses to the question about the main cause of losses indicated that, for many municipalities a problem much more important than floods caused by rivers was the various forms of flooding caused by rain, which generated significant, and likely more frequent, flood losses.

## 2.2. Data on Rainfall Floods in the Research Area

In the next phase, historical data on rainfall floods which occurred in Poland from 2010 to 2018 were analysed. Data on the locations of rainfall floods and their effects are scattered between various entities in Poland; unfortunately, there is no uniform database on these events at the national scale. The Supplementary source of information on floods caused by torrential rains was the database of interventions of the State Fire Service (Figures S1–S4, in Supplementary Materials). Interventions are undertaken in the event of a flooding occurrence and, so, information about their location provided significant support in the verification of areas at risk.

In order to illustrate the spatial extent of the effects of high-intensity rainfall, a map of rainfall flooding density was developed. Figure 2 shows the spatial differentiation of the occurrence of rainfall floods in Poland. The source of data relating to precipitation and its consequences was the sudden local floods and high-efficiency precipitation information layer developed in the GIS standard as part of the project implemented by the Institute of Meteorology and Water Management-National Research Institute, titled "Impact of the climate change on the environment, economy and society" (changes, impacts, ways of limitation, proposals for science, engineering in practice and economy planning) [22]. High-intensity rainfall was defined as stormy rainfall with a height of at least 20 mm, lasting no more than 12 h, and which resulted in local flooding, or flooding causing specific damage to environmental infrastructure and the economy.

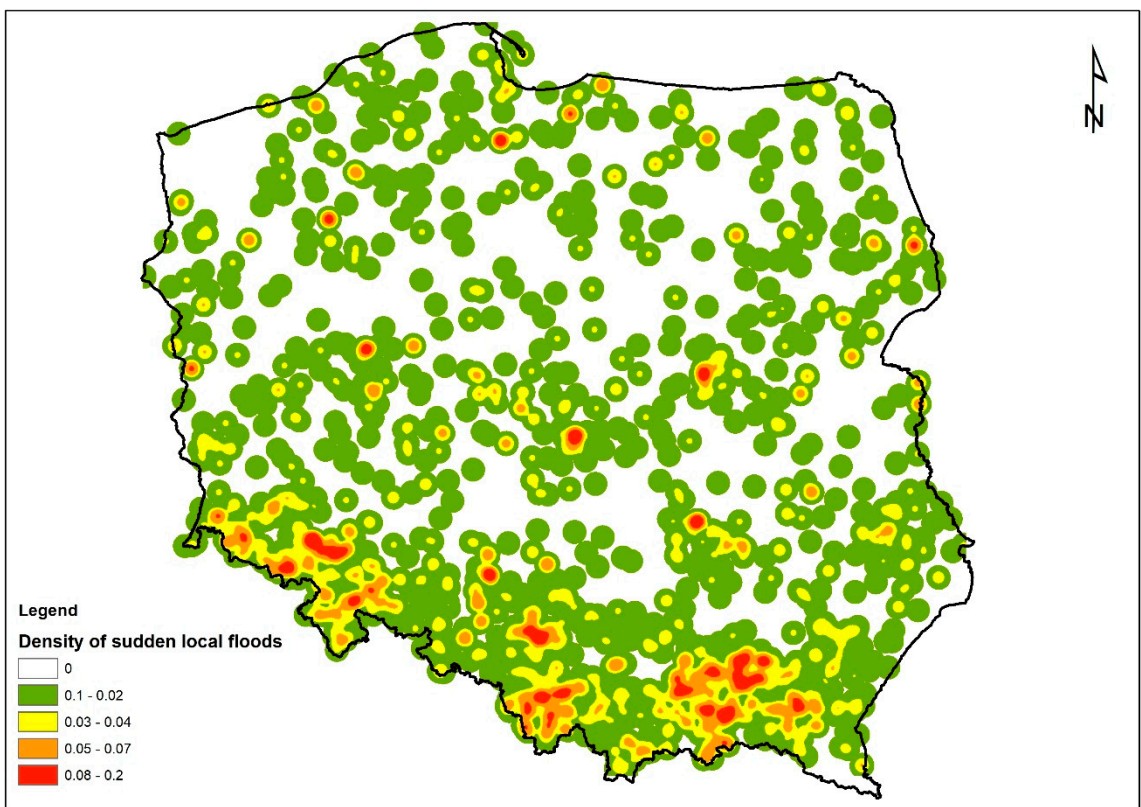

**Figure 2.** The spatial differentiation of the occurrence of rainfall floods as a result of high-efficiency precipitation. Source: Prepared on the basis of the results of the project [22].

The method of nuclear density function estimation-(Kernel Density) [23] was used to map the locations of floods for spatial analyses. This method has been implemented in one of the tools offered

by ESRI's software in the ArcGIS package. The function input parameters used in the Kernel Density (ArcGIS) tool used in the analysis were the following:

- location (x, y) of sudden rainfall floods in shape format;
- grid size of 100 m,
- for the search radius(i.e., the smoothing coefficient) preliminary analyses were performed using the radius value R = 10,000 m; and
- area unit of km$^2$.

The results of the analysis indicated that rainfall floods prevailed in the southern and south-western parts of Poland. However, dispersed single hotspots can be observed in various different parts of the country.

In conclusion, the occurrence of rainfall floods in urbanized areas is a significant problem in Poland and, so, the identification of particularly endangered areas may be important for planning, crisis management and taking corrective actions.

### 2.3. Analysis Framework for Methodology Development

In the event of a flood, risk management can be supported by information on precipitation, characterized by the randomness of occurrence, the morphology of the area, and the degree of its sealing. Due to the aforementioned randomness of precipitation, the risk assessment (in contrast to river floods) can be performed without taking into account meteorological and hydrological factors. The focus herein, thus, is geomorphological analyses relating to the topography. Factors influencing the level of flood hazard in urbanized areas include the following:

- the existence of areas without outflow;
- the existence of flat areas with a low slope of 2–3%, from which the water drains away so slowly that its quantity may be hazardous;
- the existence of areas with large terrain height differences; and
- sealing of the area, which causes rainwater to have difficult or no infiltration. Sealing is one of the most important factors influencing the accumulation of water during heavy rainfalls.

As a rule, these are largely dispersed areas, but they can also be treated as an approximate area of rainfall floods, which is more susceptible to flooding than other areas of the city. GIS tools are useful for their determination. The use of GIS in geomorphological research and soil science is often related to the analyses carried out on DTM. In the case of a DTM, GIS can be used to perform a morphometric analysis (size, shape and spatial distribution) of various forms of land relief.

### 2.4. Study Area for the Implementation of the Methodology

The city of Gdansk is interesting from the point of view of geomorphological analyses for the identification of areas at risk of rainfall flooding, as well as the practical implementation of the proposed methodology for the implementation of this task. Two examples of rainfall floods that occurred in Gdansk-in July 2001 and July 2016 -are detailed below.

Gdansk, July 2001

On July 9, 2001, a heavy rainfall occurred over Gdansk, causing a catastrophic urban flood covering the part of rivers and canals known as the Gdansk Water Junction. Intense torrential rainfall was recorded at meteorological measuring stations. Within 4 h, the rainfall was 80 mm and its value exceeded the monthly norm for July of 68 mm (the average annual rainfall for Gdansk is about 600 mm). At both stations, the rainfall on July 9 was estimated as having a probability of less than 0.3%. The daily sum of precipitation ranged from 110 mm in the northern part of Gdansk to 120 mm in the southern part.

Gdansk, July 2016

Another large flood took place 15 years later. Within 14 h, 160 mm of rainfall fell, corresponding to the two-month rainfall norm for the region. The highest rainfall occurred on the so-called the upper terrace of Gdansk, thus transforming streams into rushing rivers.

Although Gdansk is not located in the area of Poland with the highest density of rainfall floods, the terrain on which it is located favors their formation. This was confirmed by the recorded episodes of rainfall floods. The data serving as the basis for the analyses came from the digital record of interventions by fire brigades. It was assumed that any intervention by the fire brigade caused directly by rainfall was a local flood or flooding. On this basis, a map showing the cumulative number of interventions by fire brigades from 2010–2017 was developed. Figure 3 shows the cumulative value of interventions by the fire brigade in the years 2010–2017 due to heavy rainfall. The darker the color, the greater the frequency of intervention in a given place. It is also worth noting that the highest concentration of interventions was in the lower terrace of Gdansk. This is not only the most densely built-up area, but is also a vast, flat area that receives water from the upper terrace during heavy rainfall.

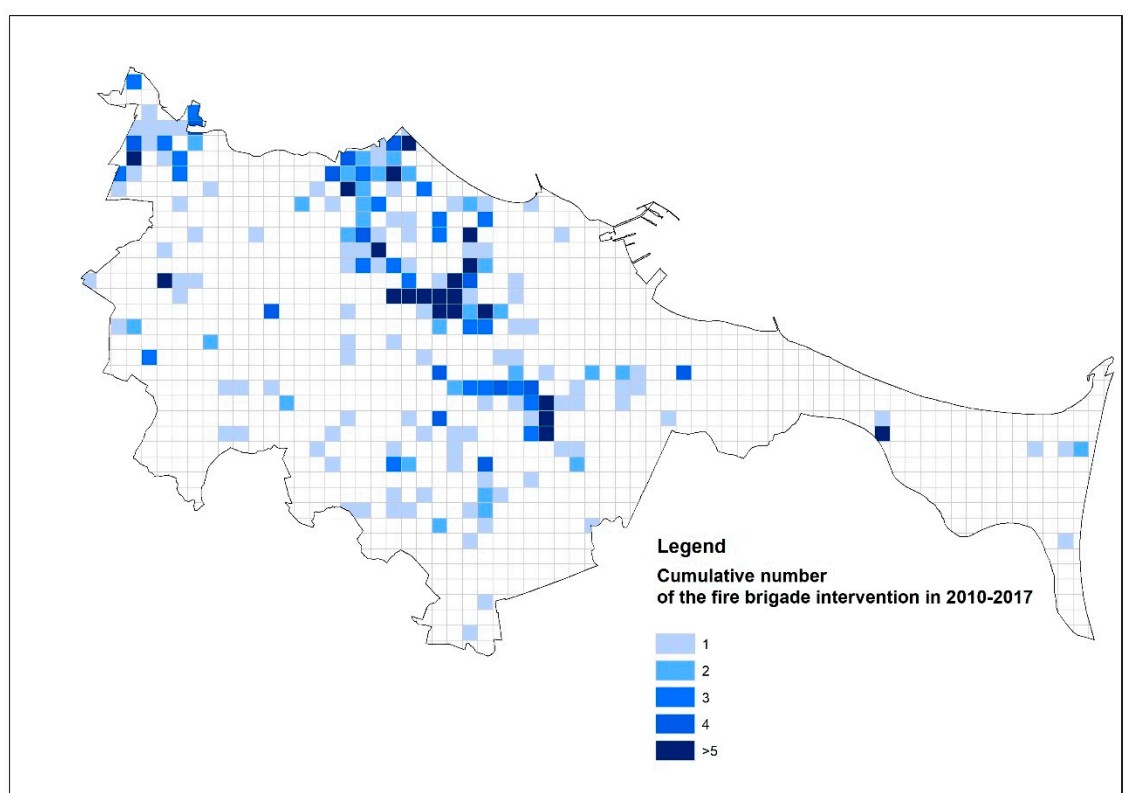

**Figure 3.** Cumulative number of fire brigade interventions in Gdansk, 2010–2017.

2.4.1. Geographic Location

Gdansk is situated on the coast of the Baltic Sea, directly on the Bay of Gdansk, and at the mouth of the Motlawa River, a tributary to the Vistula River [24]. The city is located within four different physical and geographic units according to J. Kondracki's division [25]. Generally, the level of the terrain within the city of Gdansk reaches about 180 m above sea level. The levelling of the terrain within one slope (local) is much smaller, but may also exceed 50 m in several places, while local variations of 30–40 m are common. Figure 4 shows a hypsometric map of Gdansk. The location of Gdansk within such different physical and geographic units means that the city is characterized by a large spatial diversity: topography, geological structure, hydrographic system, local climate conditions and vegetation cover. This is due to the presence of slopes with a maximum slope above 30 degrees [26]. In the spatial structure of the city, one can distinguish the so-called lower terrace the upper terrace, separated by

a highly fragmented edge zone of the plateau, where the differences in relative heights reach up to 100 m [24].

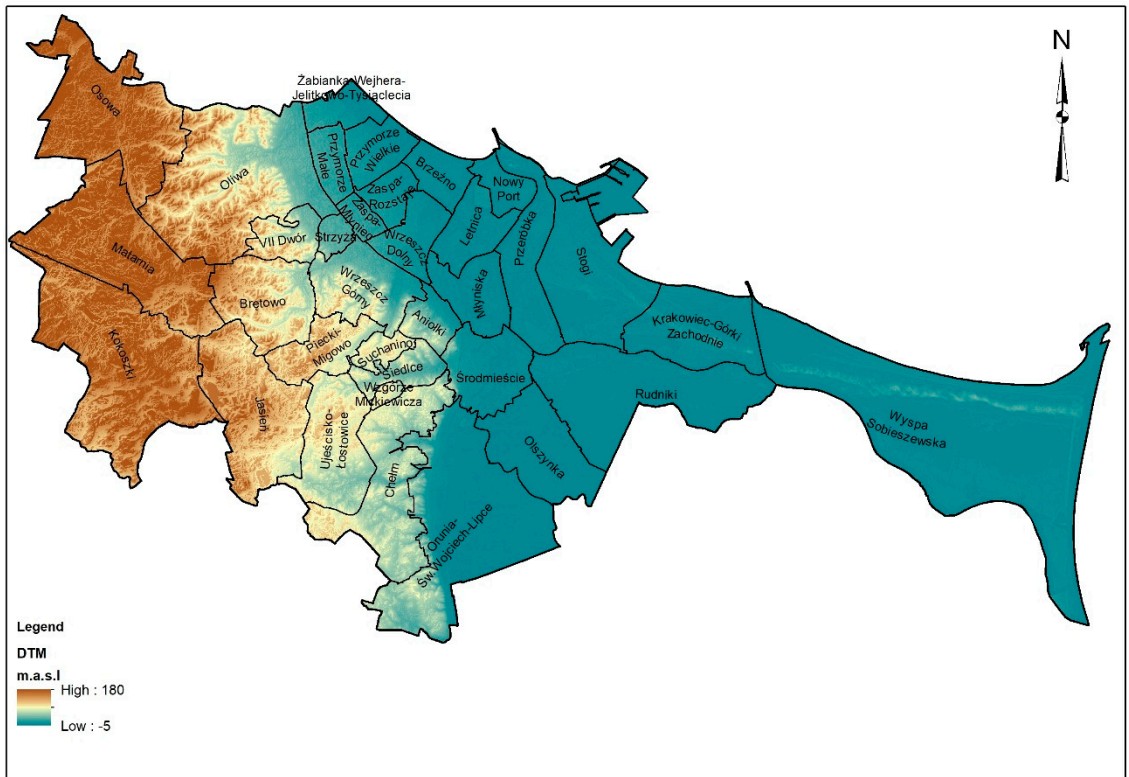

**Figure 4.** Hypsometric map of Gdansk.

### 2.4.2. Landcover

The area of the city of Gdansk is 261.3 km². Almost 52% of this area is comprised of forest, agricultural, semi-natural, and wetland areas. Industrial areas, which include public, private, and military facilities, among others, constitute 11.3% of the area. Areas with a degree of sealing above 80% constitute approximately 7.6% of the municipality's area. Figure 5 and Table 1 indicates the land-cover, according to the Urban Atlas 2012 [27].

### 2.4.3. Natural Surface Waters

Within the city limits, the surface waters include four lakes and parts of two lakes, as well as permanently or periodically wet areas of various sizes. Numerous ponds created by damming water on the watercourses flowing through the city and 53 small retention reservoirs are characteristic of the city. There are five rivers in the city, which are the main receivers of surface waters.

### 2.5. Methodology of Designating Areas at Risk of Rainfall Floods

We propose the concept of two groups of independent area predictors related to the topography and land development, defining areas favourable for the occurrence of rainfall floods. The first one includes areas related to the topography (i.e., flat areas, areas without outflow and areas with large ground height differences). A DTM of the study area was used for their identification. The second group of independent predictors are spatial development data, the source of which was the Urban Atlas [27]. In addition to the above-mentioned independent predictors, we used a database on fire brigade interventions kept by the State Fire Service [28]. This database has a data structure that is uniform on a Polish scale and is the only one that has continuously recorded information about the effects of floods. This database can be used to identify the locations of rainfall floods, map areas

potentially at risk of flooding, and assess the size of the intervention area. The proposed methodology also uses the spatial layers of water reservoirs and lakes available in the Map of Hydrographic Division of Poland (MPHP), on the scale of 1:10,000 and in the Database of Topographic Objects of Poland (BDOT) for the verification of the obtained results relating to natural water reservoirs in the study area [29].

The methodology and its intermediate results are described in detail below.

### 2.5.1. Designation of Flat Areas (First Predictor)

A DTM with a mesh resolution of 1m × 1m was used to determine flat areas (i.e., those with a slope of less than 2%). Then, resampling of the DTM grid was carried out, a raster with a resolution of 10 m × 10 m was obtained as a result of which (Please see the Appendix A). Then, its value was reclassified, in order to distinguish those areas whose decrease was less than 2%. The result of this analyses was the assessment of each area of the grid, the mesh size of which was less than 2%. Figure 6 shows the designated flat areas.

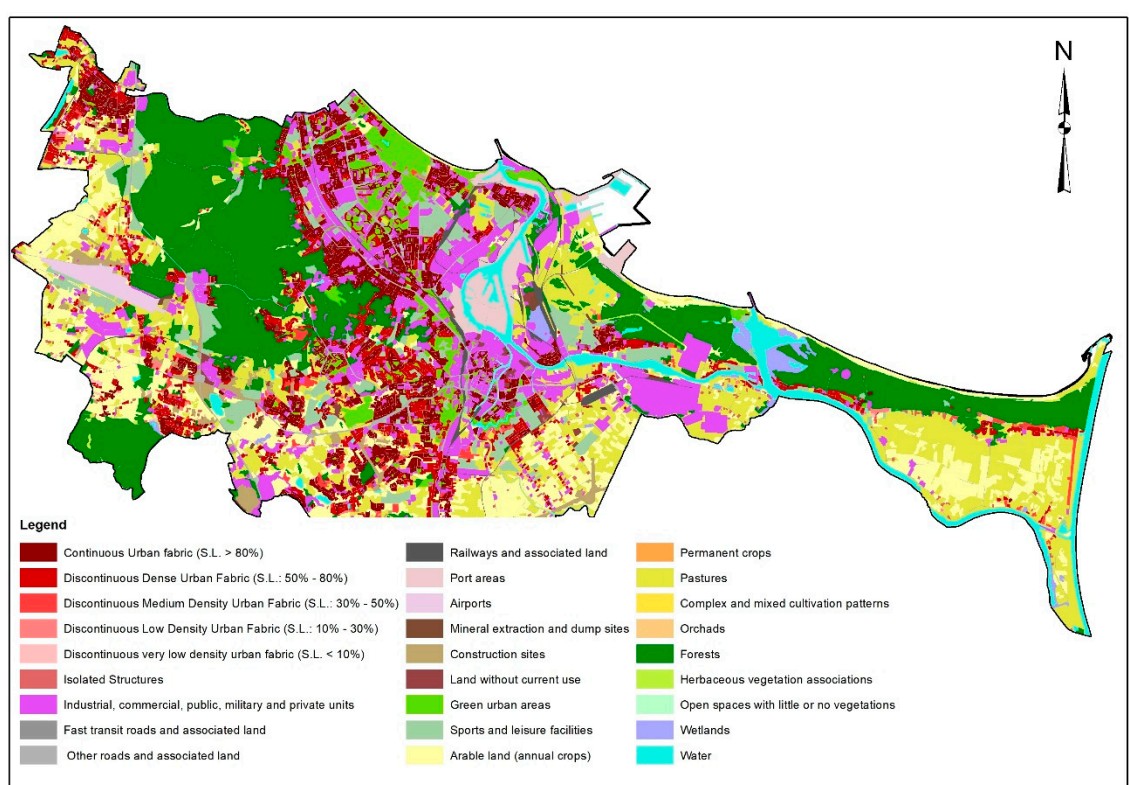

**Figure 5.** Land-cover of the area of Gdansk according to the Urban Atlas 2012 [27].

**Table 1.** Landcover in the City of GDANSK According to the Urban Atlas 2012 [27].

| Land Cover | Area [km$^2$] | Percentage of Land Cover Class in the City Area [%] |
| :---: | :---: | :---: |
| Forests | 52.9 | 20.2 |
| Arable Land (Annual Crops) | 40.2 | 15.4 |
| Pastures | 39.9 | 15.3 |
| Industrial, Commercial, Public, Military and Private Units | 29.5 | 11.3 |

**Table 1.** *Cont.*

| Land Cover | Area [km$^2$] | Percentage of Land Cover Class in the City Area [%] |
|---|---|---|
| Continuous Urban Fabric (S.L. > 80%) | 19.9 | 7.6 |
| Discontinuous Dense Urban Fabric (S.L. 50%–80%) | 13.7 | 5.3 |
| Sports and Leisure Facilities | 12.2 | 4.7 |
| Water | 11.1 | 4.2 |
| Other roads and Associated Land | 10.4 | 4.0 |
| Green Urban Areas | 8.9 | 3.4 |
| Construction Sites | 5.1 | 2.0 |
| Port Areas | 4.1 | 1.6 |
| Railways and Associated Land | 2.6 | 1.0 |
| Discontinuous Medium Density Urban Fabric (S.L. 30%–50%) | 2.4 | 0.9 |
| Wetlands | 2.1 | 0.8 |
| Airports | 1.9 | 0.7 |
| Land without Current Use | 1.3 | 0.5 |
| Isolated Structures | 1.0 | 0.4 |
| Discontinuous Low Density Urban Fabric (S.L. 10%–30%) | 0.7 | 0.3 |
| Discontinuous Very Low Density Urban Fabric (S.L. < 10%) | 0.7 | 0.3 |
| Mineral Extraction and Dump Sites | 0.5 | 0.2 |
| Total | 271.3 | 100 |

2.5.2. Designation of Areas without Outflow (Second Predictor)

A DTM with a mesh resolution 1 m × 1m (pixel type: integer) was used to determine the areas without outflow. Determination of the drainage areas consisted of the following steps:

- Filling the cavities to the specified depth Z [m] (Fill)

This operation led to the calculation of the height difference between adjacent meshes in the grid. If this difference was greater than the set value of the Z parameter, then such an area was adjusted/filled to the set value of Z. This operation was aimed at eliminating errors in the DTM. The initial value (correction) was Z = 1 m.

- Determination of flow directions for each mesh of the grid (Flow Direction)
- Designation of drainage areas (sink)

The areas without outflow were determined automatically by using the Sink tool for identification of depressions, which is available in ESRI's ArcGIS software. The analysis resulted in the development of a grid representing the driftless areas. In order to verify the designated areas, a multi-stage analysis was (carried out based on data from MPHP and BDOT) which consisted of:

- Removal of small areas without outflow.

For the criterion of a "small area", a value of the area corresponding to second percentile was adopted (p2% = 37.7), where 2% of areas had an area of less than or equal to 37.7 m$^2$ (38 m$^2$ was assumed).

- Removal of areas defined in MPHP as surface waters (i.e., rivers and lakes).
- Removal of areas defined in BDOT representing water areas and having a surface geometric representation (i.e., areas occupied by the waters of rivers, canals, water reservoirs, and other technical reservoirs).
- Removal of artifact areas resulting from DTM development. This involved the removal of areas that were, in fact, building structures. These areas were verified on the basis of BDOT data and an orthophotomap of the study area.

Figure 7 shows the designated areas without outflow.

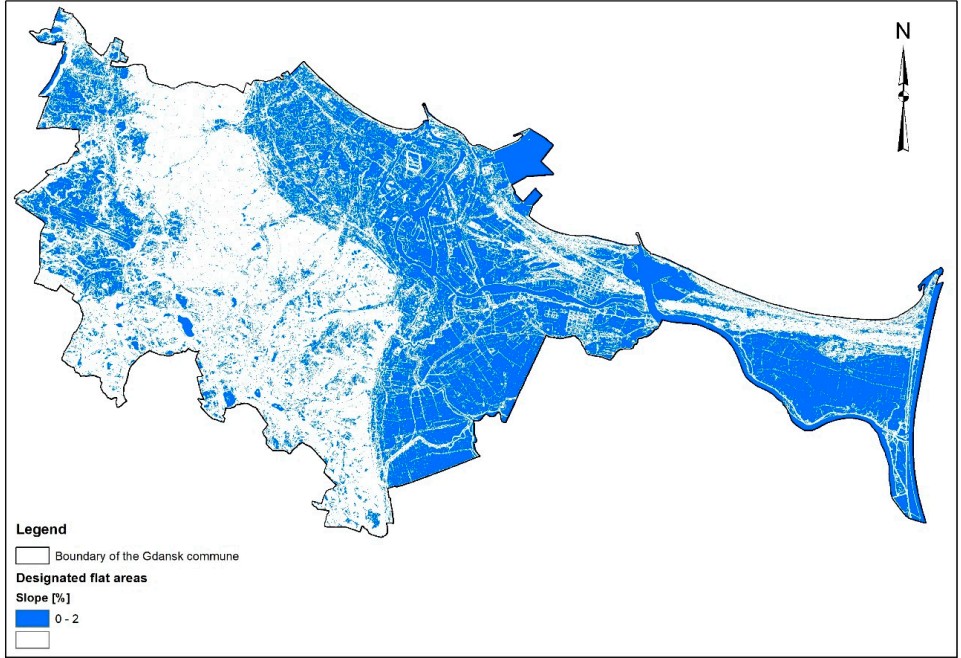

**Figure 6.** Designated flat areas of Gdansk.

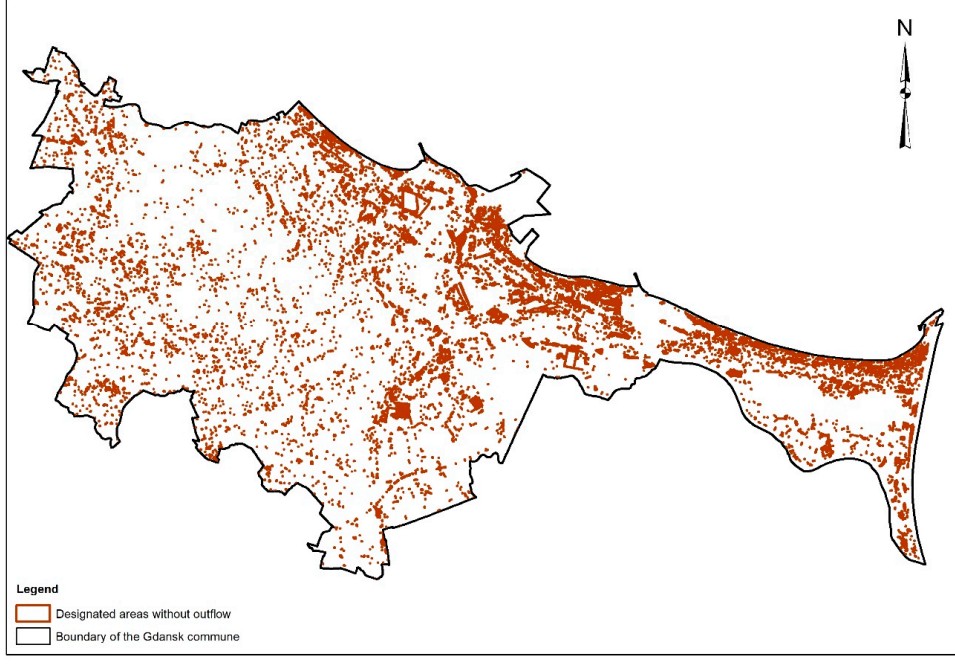

**Figure 7.** Designated areas without outflow

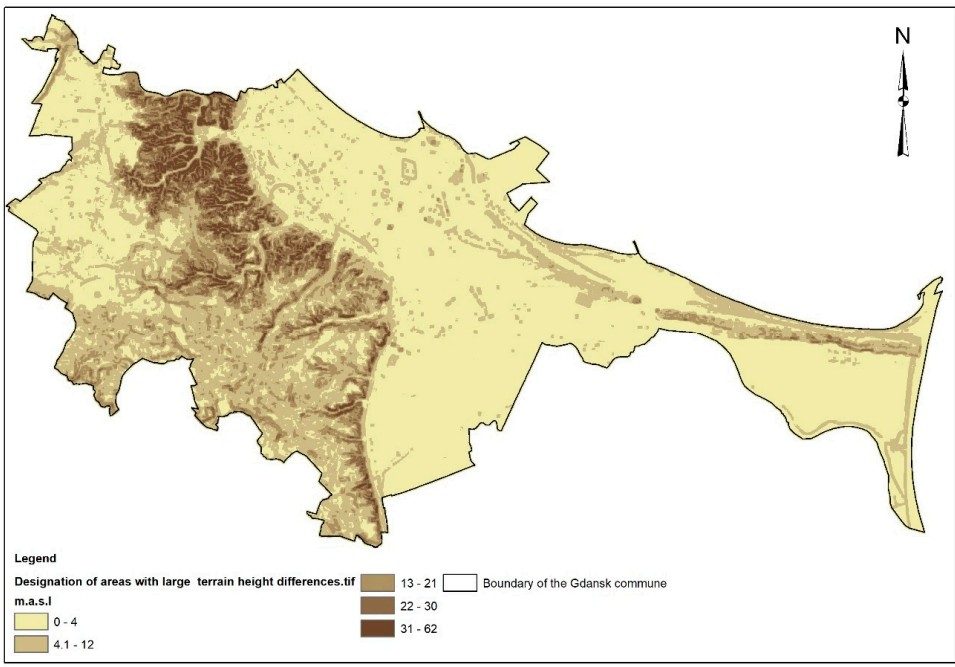

**Figure 8.** Designated areas with large terrain height differences

The end result of this part of the analysis was the development of a map of potential flood hazard areas. The map shows the spatial sum of these areas and an large terrain height differences (shown in Figure 9), an important predictor influencing the flood hazard in cities. The Geoprocessing tool, Union was used to determine these areas.

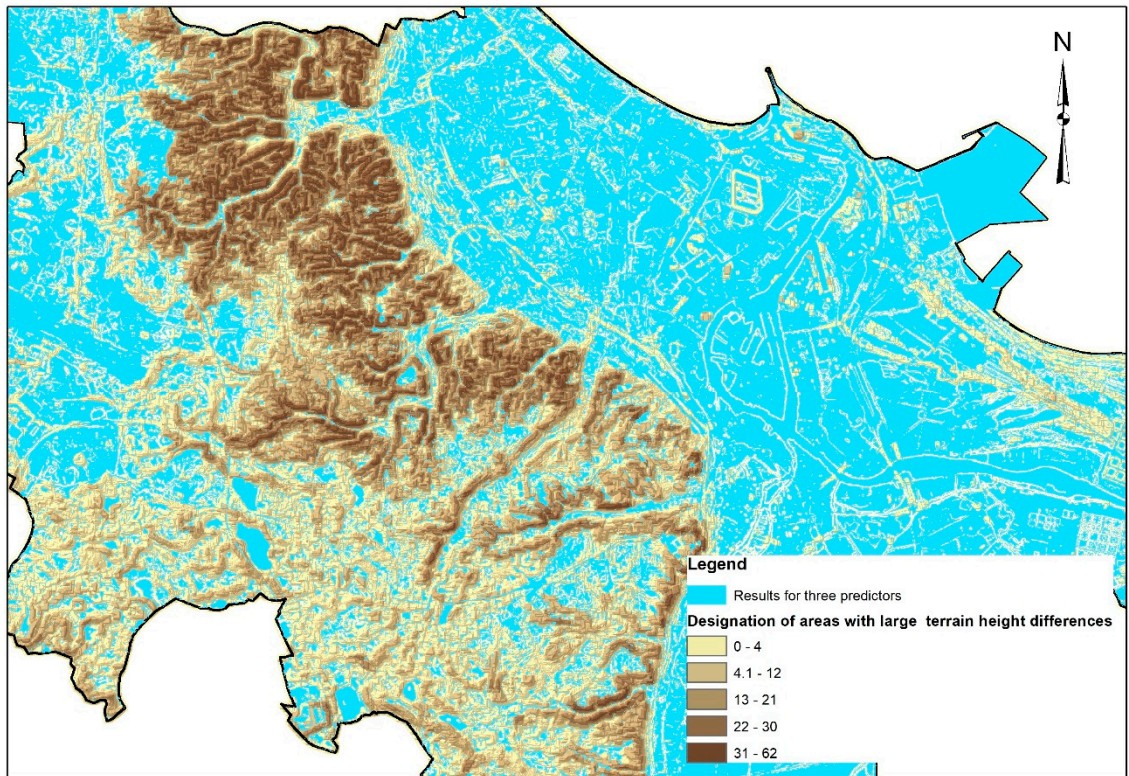

**Figure 9.** Overlaid results for three predictors

In order to better visualize the threat generated by large terrain height differences, a 3D map was developed, as shown in Figure 10. The map was developed using the Arc Scene tool and shows overlapping layers of flat areas, areas without outflow (blue), and large terrain height differences (the area highlighted as 3D). Uplifts highlighted in brown were distorted for the purpose of better visualization of the results. Figure 10 shows a clearly outlined edge zone, below which there is a rapid flattening of the terrain; this area was additionally characterized by a high degree of sealing— (see Figures 5 and 11).

### 2.5.3. Designation of Areas with Large Terrain Height Differences (Third Predictor)

A DTM with a mesh resolution of 1m × 1m (pixel type: integer), was used to determine areas with large terrain height differences. In order to determine the greatest height differences, a neighbourhood analysis (Focal Statistics) was performed. The result of this part of the analysis was the development of a grid in which the value of each cell represented the difference in height between the largest and smallest cell values in its vicinity. A square with dimensions of 100 × 100 was adopted as the shape of the neighborhood. Figure 8 shows the designated areas with large terrain height differences.

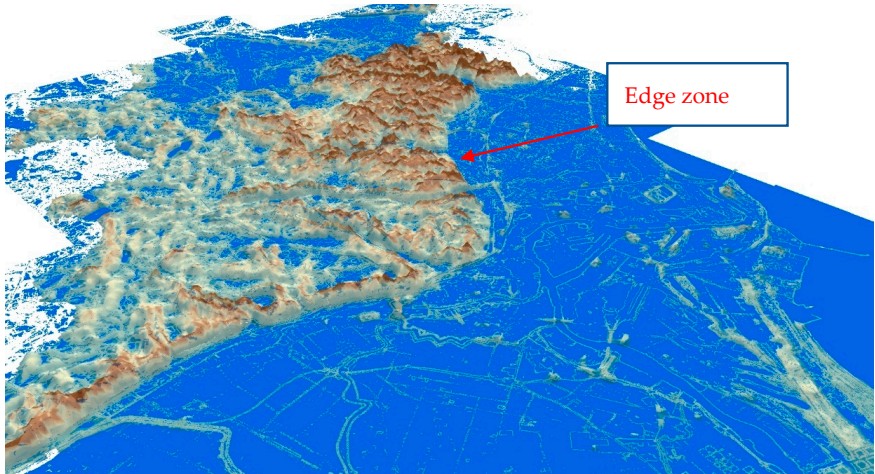

**Figure 10.** 3D view showing flat areas and areas without outflow (blue) and areas with large terrain height differences, view from the sea.

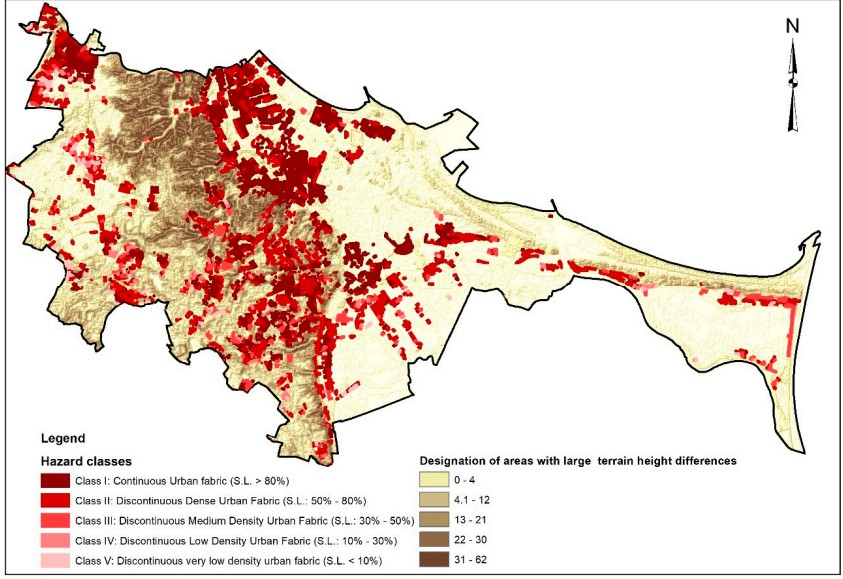

**Figure 11.** Classification of areas at risk of rainfall flooding (classes from I to V) against the background of areas with large terrain height differences.

## 3. Results

Designated areas should be prioritized to reflect the level of risk. The identification of this level is important, from the point of view of urban flood management as a part of crisis management. In this classification, the degree of sealing can be used which indicates the difficulties in the outflow of rainwater from endangered areas. The higher the degree of sealing, the longer the water layer will remain in these areas, posing a threat to the people and infrastructure facilities located in the area. The classification may, therefore, support rescue activities, as well as spatial planning (in terms of revitalization of these areas).

The classification of the areas at risk of rainfall floods was carried out using information on the degree of land sealing from the Urban Atlas 2012 database (fourth predictor).

- The first step in the analysis was to separate the five classes from the Urban Atlas 2012 database characterized by the highest degree of land sealing:

  1. Continuous urban fabric (S.L. > 80%)
  2. Discontinuous dense urban fabric (S.L.: 50%–80%)
  3. Discontinuous medium-density urban fabric (S.L.: 30%–50%)
  4. Discontinuous low-density urban fabric (S.L.: 10%–30%)
  5. Discontinuous very low-density urban fabric (S.L. < 10%)

- Then, geoprocessing tools were used to perform the classification. Using the Intersect tool, the geometric common part of the areas without outflow and flat areas, as well as the five selected classes of land sealing were separated.
- The next step was to aggregate the data using the Dissolve tool. This tool made it possible to combine objects based on defined attributes containing information about the degree of land sealing.
- To eliminate void artifacts inside regions, the Buffer tool was used, which was used to compute a 50 m radius around each region.

The results of the above analyses are presented, against the background of areas with large terrain height differences, in Figure 11.

Verification of the results for the four predictors in the study area was carried out using the data on firefighting interventions.

Figure 12 shows the spatial distribution of the fire brigade interventions, along with the following predictors of precipitation floods:

1. Flat areas with low slopes,
2. Areas without outflow,
3. Areas with large terrain height differences, and
4. Spatial development land-cover classes.

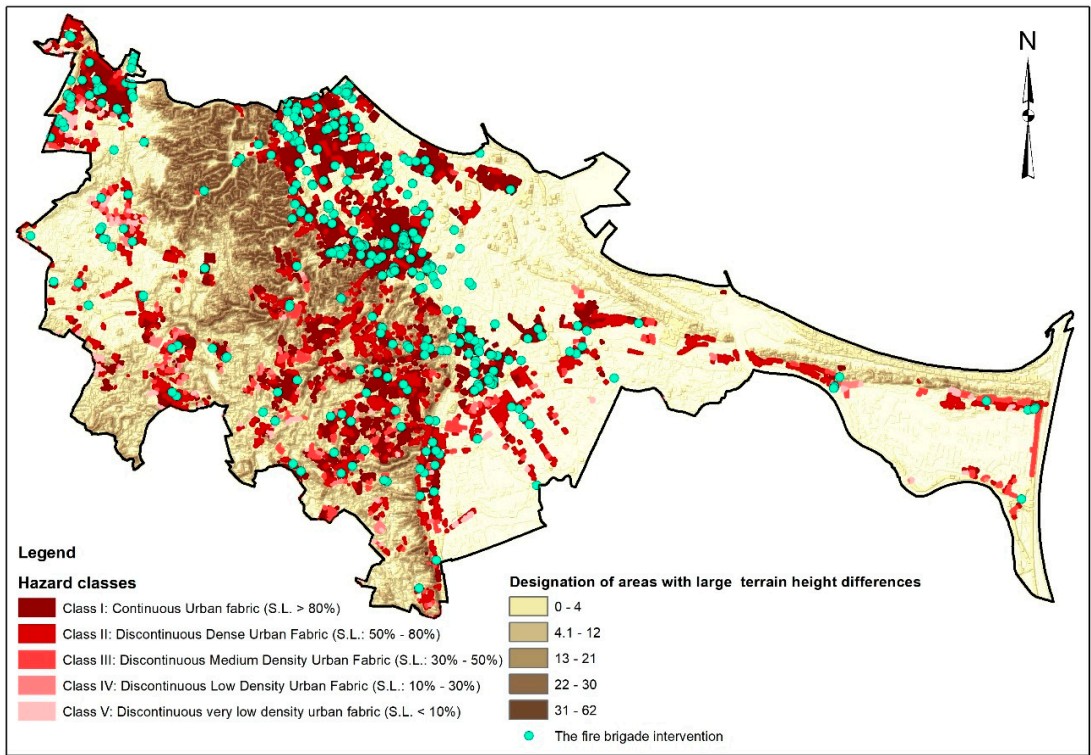

**Figure 12.** Interventions by the fire brigade against the background of the results for the four rainfall flooding predictors in the study area.

For all four predictors, there was a strong association with fire brigade intervention sites due to heavy rainfall. It can be seen that in most cases, the interventions were concentrated in the lower part of the city (i.e., on the lower terrace). This part of the city of Gdansk is characterized by a significant share of flat areas. It is worth noting that, in the areas without outflow, there were also high local height differences of 30 m, sometimes even more than 60 m. The analysis also showed that the highest concentration of fire brigade interventions occurred along the edge zone extending from the Vistula valley to the north-west (including the Coastline). These areas are characterized by large terrain height differences. Apart from the above-mentioned features, this area was characterized by the greatest sealing level.

## 4. Discussion

The extremely diverse urban infrastructure is a factor that hinders the determination of areas at risk of rainfall flooding. This is due to not only sewage systems, but also roads and buildings, which can affect the direction of rainwater flow (by blocking natural runoff; as a result, water may accumulate in depressions of the terrain or form a layer of water on flat areas), thus making it difficult to designate such areas and plan for effective management of them. The analyses presented above show that there are, however, tools and data that may allow for the designation of such areas. Predictor groups may influence the flood risk in various ways, depending significantly on the local situation. The most vulnerable places in cities are low-lying points, such as underpasses, underground garages, tunnels and basements, which can become death traps [29]. It may be assumed that, except for underground passages and tunnels, the depth of flooded streets, yards, and squares is usually no greater than 30–60 cm [30]. The analysed case study area has a particularly complex terrain morphology, thus favoursing rainfall floods in the event of sudden torrential rainfalls. This was confirmed by the numerous interventions by the fire brigade. The proposed methodology uses GIS tools to designate risk areas and is independent of rainfall data.

There are many factors and corresponding data that contribute to flooding in urbanized areas. The only question is determining the level of availability that we want to use this data from. The data on the analysed area of Gdansk used in the presented methodology are available from global databases—and, so, are, generally accessible. Therefore, it is possible to perform country-level general analyses on this basis. However, there are other factors and related data that are locally available and, in practice, require separate preparation. These include the historical system of water courses and a diagnosed malfunctioning rainwater drainage system (i.e., water flowing out of an overfilled sewage system). Taking such information into account makes sense when we prepare analyses at the city or district level; only then can we guarantee access to such data.

The aim of the proposed methodology is primarily to reduce the flood risk in designated areas through changes in spatial planning, relocation of endangered resources, revitalization of sealed areas, and limiting further development. The designation of areas also allows for the planning of rescue operations and designing of escape routes. The proposed methodology can be used in the development of a catalogue of good rainfall management practices in the city and for adaptation to climate change in Gdansk.

## 5. Conclusions

It is assumed that the proposed method can be used in urbanized areas: relatively small but intensively built-up with a high ratio of sealed surfaces, where the highest unit losses per unit of flooded area may occur. The designated areas are often characterized by a small area (i.e., not exceeding several hundred square meters) but, in this case, the most important focus was placed upon the losses in municipal and private infrastructure [31]. The main goal of the proposed method was to identify areas at risk of rainfall floods (i.e., those that do not arise as a result of flooding of rivers or flooding of coastal areas as a result of storm surges and their flooding with sea waters and backwaters of rivers). For this purpose, hydrodynamic modelling tools, such as MIKE FLOOD [32] were used. Hydrodynamic models were used for this area, in the same way as for the whole of Poland for the development of flood hazard maps from river floods. Referring to the location of Gdansk, as a pilot area located on the coast, scenarios developed for the period 2081–2100 have indicated [22] that the average annual sea level will increase significantly, compared to the reference period adopted in the project (1971–1990). This increase in the average annual sea level of the eastern part of the Polish coast is estimated to be approximately 0.35 m. The proposed method, therefore, may be used to identify flat and non-drainage areas at risk of flooding as a result of sea level rise without requiring hydraulic modelling [33]. Hydrodynamic models were not used to determine the areas at risk of rainfall floods in the pilot site; instead, the topography and the degree of its sealing were analysed. Areas at risk of rainfall flooding were designated using GIS tools offered by ArcGIS. The use of these tools does not require additional data, such as soil moisture content. In the case of urban flooding caused by short duration torrential, the initial degree of soil water saturation is negligible. Analyses performed on the basis of the proposed methodology do not require additional tools in the field of water management. Thanks to this, they can be carried out by municipal planning offices. Of course, there also exist professional modelling solutions. An example of a rainwater management model is the U.S. Environmental Protection Agency's Storm Water Management Model (SWMM) [34]. This is one of the programs that facilitates the design process, which can evaluate the operations of rainwater drainage systems. The mathematical model allows the amount of precipitation to be transposed into a run-off which fills pre-determined elements, such as a system of pipes, open channels, pumping stations, storage reservoirs, and sewer outlets. It is also possible to analyse related parameters (e.g., sewage filling height, wastewater volumetric output at any point in the network, wastewater quality indicators, and so on).

However, simulation carried out using the EPA SWMM program can only produce reliable results with an adequate database of rainfall, catchment parameters, sewage networks, and/or soil characteristics. Modelling in SWMM must be supported by calibration of the parameters set by default

by the program's algorithm; in turn, correctly performed calibration must be supported by data on the relationship between rainfall and runoff.

In this paper, the fire brigade intervention database was used as a supplement, for the proposed methodology. However, it should be emphasized that fire brigade interventions may be the subject of separate analyses related to the study of the correlation between fire brigade intervention sites and the sealing of the surface of the city of Gdansk.

Practical use of the proposed methodology may also be related to the crisis management cycle. Crisis management consists of the following phases: early warning, emergency, response, reconstruction, prevention and preparedness. In the case of early warning, accurate prediction of the place of precipitation and issuing an appropriate warning may concern larger areas exceeding the city limits [35]. The areas designated using on the proposed methodology may allow for the proper planning of rescue actions in the response phase. As far as prevention is concerned, important actions should concern spatial planning, reducing the degree of sealing in hazardous areas, and designing adequate drainage systems. The designation of risk areas based on the proposed methodology may have a significant impact on the preparedness phase. Readiness consists of knowledge, including knowledge of the threat as well as the actions that can be taken to prevent, react, and mitigate its potential consequences. Knowledge is transferred through information. This distinction is important: Information is knowledge-based, but whether information is received as intended will depend on the existing knowledge of the audience. The preparation of appropriate information, in the form of messages and leaflets regarding areas at risk of flooding and directing them to interested parties, can significantly reduce potential losses. Their limitation may be made possible by including, in the information provided, recommendations regarding measures to prepare for possible flooding. Such activities should include securing cellars and moving valuable items to higher places in endangered facilities.

**Supplementary Materials:** The following are available online at http://www.mdpi.com/2076-3263/10/11/457/s1, Figure S1. "Manhattan" in Gdansk according to the analysis in DTM 1 m x 1 m. Figure S2. ID number before dissolve function (3 separate polygon objects: 117, 118, 118), but with the same GRIDCODE (82). Figure S3. Example of object after the tool was applied. Figure S4. 2% of the areas had an area less than or equal to 37.7 m$^2$; 38 m$^2$ was assumed for the GIS analyses.

**Author Contributions:** Concept of the article by T.W. and M.S. Data collection and analysis within the doctoral dissertation M.S. Methodology, T.W. and M.S. Software, M.S. Validation, T.W. and M.S. Formal Analysis and Investigation, T.W. and M.S. Resources, M.S. Writing and original draft preparation, T.W. Writing, review and editing, T.W. Visualization, M.S. Supervision, T.W. Project administration, T.W. All authors have read and agreed to the published version of the manuscript.

**Funding:** This research received no external funding.

**Acknowledgments:** The presented research works were carried out by the authors at the Institute of Meteorology and Water Management-National Research Institute. The source of research and publication funding is a subsidy from the Minister of Science and Higher Education allocated to statutory research activities. In addition, the results of the project "Impact of the climate change on the environment, economy and society (changes, impacts, ways of limitation, proposals for science, engineering in practice and economy planning)" were used (the grant agreement No.POIG.01.03. 01-14-011 / 08 - 00 of December 1, 2008 as part of the Innovative Economy Operational Program, Project 1, Measure 1.3, Sub-measure 1.3.1). The results of research for National Water Management Board carried out by the authors in the process of implementing the Floods Directive in Poland were also used.

**Conflicts of Interest:** The authors declare no conflict of interest.

## Appendix A

Determination of sink areas based on the DTM integer type, resolution 1 m × 1 m.

Comments

(1) Changing the raster type nmt_gd from floating point to integer type, new raster name—nmt_gd_int

(2) Postprocessing, i.e. filling cavities up to 1m using the fill tool, a new raster called Fill_nmt_gd_i

(3) Flow direction on the raster named Fill_nmt_gd_i, a new raster named fd_nmt_i was created

(4) Generating a raster called Sink_nmt_i with drip areas - sink. As a result of this operation, the raster numbered 85,869 "objects".

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
