# Peer review of "Rainfall Flooding in Urban Areas in the Context of Geomorphological Aspects"

_geosciences, doi:10.3390/geosciences10110457_

Round 1

Reviewer 1 Report

The study is about the identification of flood risk area considering geomorphological information. However, there was no result or method to develop the risk map of the city. Only input data were shown in the figures. Therefore, it should be revised very much to be published in the journal.

2.1 and 2.2 are not necessary to explain the studied area. Too detailed explanation about the whole area of the Poland.

Results just show the geological information about the studied area. Please describe the detailed method how to develop the flood risk area considering all of the information.

Discussion or conclusions should be rewritten to explain the results found in this study.

Author Response

Dear Reviewer,

Thank you for your input. Please  find our answers and explanations in the attachment.

Reviewer 2 Report

Please see the attached document

Author Response

(The authors gave the same response as above.)

Round 2

Reviewer 1 Report

The manuscript were revised very well as the comments previously suggested. Therefore, it could be considered as the possible publication in the journal.

Author Response

Dear Reviewer,

Thank you for your input. Please see our additions.

Best regards,

Authors

Reviewer 2 Report

The authors have revised the Paper to meet the (main) requests for change and the suggestions I recommended.
I feel that the review process led to an improved manuscript, and I suggest the acceptance of the paper for publication in Geosciences.

Sincerely Yours,

Author Response

Dear Reviewer,

Thank you for your input. Please find in the attachment our additions to the Introduction and Results.

Best regards,

Authors
